

# Increased academic stress is associated with decreased plasma BDNF in Chilean college students

Juan-Luis Castillo-Navarrete[1,2,3], Claudio Bustos[2,3,4], Alejandra Guzman-Castillo[2,3,5] and Benjamin Vicente[2,3,6]

[1] Departamento Tecnología Médica, Facultad de Medicina, Universidad de Concepción, Concepción, Chile
[2] PhD Programme in Mental Health, Facultad de Medicina, Universidad de Concepción, Concepción, Chile
[3] Programa de Neurociencia, Psiquiatría y Salud Mental, NEPSAM, Universidad de Concepción, Concepción, Chile
[4] Departamento de Psicología, Facultad de Ciencias Sociales, Universidad de Concepción, Chile, Chile
[5] Departamento de Ciencias Básicas y Morfología, Facultad de Medicina, Universidad Católica de la Santísima Concepción, Concepción, Chile
[6] Departamento de Psiquiatría y Salud Mental, Facultad de Medicina, Universidad de Concepción, Concepción, Chile

Corresponding author
Juan-Luis Castillo-Navarrete,
jucastillo@udec.cl

## ABSTRACT

**Introduction**. Academic stress (AS) is a prevalent challenge faced by university students, potentially affecting molecular indicators such as brain-derived neurotrophic factor (BDNF) and global DNA methylation (G-DNA-M). These indicators could illuminate the physiological ramifications of academic stress.

**Study Design and Methods**. This research followed a quantitative, non-experimental, longitudinal panel design spanning two academic semesters, observing phenomena in their natural context. Students from the Medical Technology program at Universidad de Concepción, Chile were involved, with assessments at the beginning and during heightened academic stress periods.

**Sample**. Of the total participants, 63.0% were females, with an average age of 21.14 years at baseline, and 36.92% were males, averaging 21.36 years. By the study's conclusion, female participants averaged 21.95 years, and males 22.13 years.

**Results**. Significant differences were observed between initial and final assessments for the SISCO-II Inventory of Academic Stress and Beck Depression Inventory-II, notably in stressor scores, and physical, and psychological reactions. Gender differences emerged in the final physical and psychological reactions. No significant changes were detected between the two assessments in plasma BDNF or G-DNA-M values. A refined predictive model showcased that, on average, there was a 3.56% decrease in females' plasma BDNF at the final assessment and a 17.14% decrease in males. In the sample, the G-DNA-M percentage at the final assessment increased by 15.06% from the baseline for females and 18.96% for males.

**Conclusions**. The study underscores the physiological impact of academic stress on university students, evidenced by changes in markers like BDNF and G-DNA-M. These findings offer an in-depth understanding of the intricate mechanisms regulating academic stress responses and highlight the need for interventions tailored to mitigate its physiological and psychological effects.

## INTRODUCTION

Stress is an organism's multifaceted response to any demand (*Selye, 1956*). The initiating stressor prompts either an acute or chronic reaction (*Selye, 1973*; *Tafet, 2022*). Acute stress elevates heart rate, blood pressure, respiratory rate, glycemia, and coagulation factors (*Tafet & Nemeroff, 2016*; *Tafet & Nemeroff, 2020*). This leads to a coping phase, then relaxation, and finally, physiological homeostasis (*Tafet, 2008*; *Tafet & Nemeroff, 2016*; *Tafet & Nemeroff, 2020*). Chronic stress begins with an alert phase, then a resistance stage. An exhaustion phase follows, stemming from prolonged stressor exposure, disrupting homeostasis. This imbalance triggers pathophysiological events like increased proinflammatory cytokines, hippocampal structural changes, and hormonal shifts (*Conrad, 2010*; *Hänsel et al., 2010*; *Nargund, 2015*; *Sribanditmongkol et al., 2015*). Furthermore, chronic stress is linked to cardiovascular disease and cancer development or prognosis (*DeSteno, Gross & Kubzansky, 2013*; *Dhabhar, 2014*; *Steptoe & Kivimäki, 2012*; *Tian et al., 2014*).

Academic stress (AS) arises from educational experiences. It starts in primary students and intensifies through university (*Dyson & Renk, 2006*; *Putwain, 2007*). Higher education epitomizes this stress due to hefty workloads. Concurrently, students often face family separation, job entry, and new environments (*Barraza-Macías, 2007*; *Castillo-Navarrete et al., 2023a*). University students encounter notably stressful periods, requiring significant adaptation. This can lead to exhaustion, dwindling study interest, and environmental challenges (*Guzmán-Castillo et al., 2022*).

Brain-derived neurotrophic factor (BDNF) is a protein prevalent in the adult brain. It's found in most cortical areas, several subcortical regions, and spinal cord sections. BDNF is vital for functions like dendritic branching, spine morphology, synaptic plasticity, and long-term potentiation (LTP). Consequently, it affects memory, learning, appetite, and sleep (*Castillo-Navarrete et al., 2023a*; *Karege et al., 2002*; *Lommatzsch et al., 2005*; *Nagahara & Tuszynski, 2011*).

The BDNF gene, on 11p13, features a nonconservative exonic SNP at nucleotide 196 (dbSNP rs6265, G/A). This SNP transforms valine to methionine in proBDNF 5'protein at codon 66 (Val66Met). This change impacts BDNF packaging, reducing its release-dependent activity. The Val/Met variant correlates with cognitive deficits, impaired memory, and reduced hippocampal activity (*Nagahara & Tuszynski, 2011*; *Vyas & Basant, 2012*).

BDNF in peripheral blood (PB) originates from platelets and the brain (*Lommatzsch et al., 2005*; *Polacchini et al., 2018*; *Suliman, Hemmings & Seedat, 2013*). Since it traverses the blood–brain barrier, peripheral BDNF indicates brain BDNF levels and cortical health (*Karege et al., 2005*; *Polacchini et al., 2015*; *Suliman, Hemmings & Seedat, 2013*). BDNF

levels in PB are measured in serum, plasma, or platelets (*Castillo-Navarrete et al., 2023a*; *Polacchini et al., 2015*; *Teche et al., 2013*).

Stress profoundly impacts BDNF (*Bath, Schilit & Lee, 2013*; *He et al., 2020*). Chronic stress activates the hypothalamic-pituitary-adrenal (HPA) axis, elevating cortisol and neuroinflammatory cytokines while reducing BDNF levels (*Brunoni, Lopes & Fregni, 2008*; *Castillo-Navarrete et al., 2023a*). Assessing peripheral BDNF during acute stress can help gauge HPA axis activity (*Castillo-Navarrete et al., 2023a*; *Linz et al., 2019*; *Schmitt, Holsboer-Trachsler & Eckert, 2016*; *Schulte-Herbrüggen et al., 2006*).

BDNF supports neuroplasticity and brain health, with its expression affected by stress types, including chronic, acute, and post-traumatic (*Aksu et al., 2018*; *Martinotti et al., 2016*; *Zamani et al., 2019*). Psychiatric disorders like depression and schizophrenia exhibit altered BDNF levels (*Martinotti et al., 2016*; *Suliman, Hemmings & Seedat, 2013*; *Vinogradov et al., 2009*). Cognitive training in schizophrenics elevates serum BDNF (*Martinotti et al., 2012*; *Vinogradov et al., 2009*). Therapies, including antidepressants and psychotherapy, also boost BDNF (*Kishino et al., 2001*; *Martinotti et al., 2016*; *Vanicek et al., 2019*). These findings highlight BDNF's role in stress-related psychiatric disorders and therapeutic modulation potential.

Traumatic experiences may predispose individuals to stress disorders through epigenetic modulation (*Sant & Goodrich, 2018*). This can involve DNA methylation, histone changes, and non-coding RNA processes, influencing gene transcription (*Hing, Sathyaputri & Potash, 2018*). 5-methylcytosine (5-mC) is a methyl group attached to cytosine's carbon 5. Global DNA methylation (G-DNA-M) measures 5-mC levels against total cytosine (*Murgatroyd, 2014*; *Paul & Tollefsbol, 2014*; *Yamamoto et al., 2014*). While many studies target specific genes, G-DNA-M's role, present in coding and non-coding areas, is noteworthy (*Makhathini et al., 2017*).

University students experience distinct stress levels, possibly causing epigenetic shifts affecting BDNF gene expression. Despite its importance, few studies explore AS's impact on peripheral BDNF and G-DNA-M percentages. This research seeks to discern if AS affects plasma BDNF and global DNA methylation percentages, offering insights into academic stress's intricate regulatory mechanisms.

# METHODS

## Methods

### Design

This is a quantitative, pilot research with a non-experimental, longitudinal panel design. It involves initial and final evaluations. The study is non-experimental, observing phenomena in their natural context without variable manipulation. It's longitudinal, collecting data over two academic semesters at two intervals. Being a panel study, the same group was assessed throughout (*Hernandez Sampieri, Fernandez Collado & Baptista Lucio, 2010*).

### Participants and procedure

Students from the Medical Technology (MT) program at Universidad de Concepción (UdeC) participated. Through concise, voluntary sessions, II to V-year students from

UdeC's MT program were invited. Interested participants received details about the study's objectives, their involvement, and the minimal risks, primarily from standard phlebotomy. In the initial stress phase, during the first 10 days of the 2018 semester, 91 students participated, providing blood samples, and completing psychometric evaluations. The subsequent sampling, during the heightened academic stress period, occurred between December 2018 and January 2019. Here, blood samples were retaken, and appropriate evaluations were conducted.

### Ethical considerations

The authors confirm adherence to ethical standards of pertinent national and institutional human experimentation committees, aligning with the 1975 Helsinki Declaration, updated in 2008. The Scientific Ethical Committee of the Faculty of Medicine at Universidad de Concepción approved all human-related procedures (No CEC 65/2018). Every participant provided signed informed consent, ensuring complete anonymity within the manuscript. No reward was offered for participation.

### Exclusion criteria

Participants without signed informed consent were excluded. Additionally, those receiving pharmacological or psychological treatment for mental health issues were omitted.

## Instruments

### SISCO-II inventory of academic stress (SISCO-II-AS)

This Spanish instrument comprises 33 items. Eight items gauge the perceived stress from environmental demands. The "total reaction" section (17 items) assesses physical and psychological reactions (11 items) and social behavioural reactions (six items). The final six items evaluate the frequency of employed coping strategies (*Castillo-Navarrete et al., 2020*; *Guzmán-Castillo et al., 2022*).

### Beck depression inventory-II

This 21-item self-report tool evaluates depressive symptom severity in adults and adolescents aged 13 and up. Scores range from 0 to 63 (*Melipillán et al., 2007*). Both Beck and Dozois pinpointed a 19-score threshold on the BDI-II to identify significant depressive symptoms (*Dozois, Dobson & Ahnberg, 1998*). Beck's established BDI-II thresholds are 0–13: minimal; 14–19: mild; 20–28: moderate; and 29–63: severe (*Beck, Steer & Brown, 1996*).

### Alcohol use disorders identification test (AUDIT)

This tool, created by the World Health Organization, screens for excessive alcohol consumption and aids in brief interventions (*Donoso, 2015*). Comprising 10 items (up to 40 points), it spans three domains: risk drinking, dependence symptoms, and harmful drinking. Thresholds are set at (i) 6–8 items for risk drinking and (ii) 9 or more items for harmful drinking or dependence (*Alvarado et al., 2009*; *Donoso, 2015*).

### Self-report symptom questionnaire (SRQ20)

This nonspecific scale evaluates somatic complaints, global functioning, anxiety, and depressive symptoms, with a top score of 20 points. It differentiates only between present

and absent symptoms. Scores exceeding 8 indicate a likely high risk of a mental disorder (*Illanes et al., 2007*; *Vicente et al., 1994*).

## Sampling and laboratory procedure
### Peripheral blood collection
At both basal and final stress stages, 12 ml of peripheral blood was drawn from each participant, using tubes with EDTA-K3. Post-collection, tubes underwent centrifugation at 500 g for 10 min. Subsequently, the required components (plasma and buffy coat) were isolated, aliquoted, and stored at −80 °C.

## DNA extraction process
From the preserved buffy coat, DNA was extracted utilizing the "Blood DNA Preparation Kit" from Jena Bioscience (PP205S) (Jena Bioscience, Jena, Germany). The Infinite® 200 PRO NanoQuant assessed the DNA's concentration and purity. The extracted DNA was then stored at −80 °C until needed.

## Plasma BDNF measurement
The "Human BDNF PicoKine™ ELISA Kit" by Boster (EK0307) (Boster Bio, Pleasanton, CA, USA) was employed. This solid-phase immunoassay is tailored for human BDNF measurement. Each plate incorporated a calibration curve ranging from 0.025 pg/ml to 2.116 pg/ml. The Infinite® 200 PRO NanoQuant device recorded absorbance at 450 nm. All tests were conducted in duplicate, with results presented in pg/ml.

## Assessment of global methylation percentage
Using stored DNA, the "MethyFlash™ Global DNA Methylation (5-mC) ELISA Easy Colorimetric" kit (Epigentek, P-1030) was employed. Absorbance was recorded at 450 nm using the Infinite® 200 PRO NanoQuant device. Each determination was conducted in duplicate, using a minimum of 100 ng of DNA per well. A calibration curve, spanning 0%–5% total methylation, aided in calculating the methylated DNA percentage. For every sample, the formula incorporated the DNA concentration, sample absorbance, negative control absorbance, and the calibration curve's slope:

5mc% = [(Sample absorbance) −(Negative control absorbance)/(curve slope)x(DNA concentration in the well)] × 100.

## Val66Met polymorphism analysis
The allele-specific PCR method was employed to study the Val66Met polymorphism. Specific primers were crafted to discern each allele type (either homozygous or heterozygous) through conventional PCR (*Furuta et al., 2007*).

## Statistical evaluation
### Data source
Access the utilized data at https://doi.org/10.48665/udec/ORVXAA.

### Descriptive and bivariate analysis
An initial descriptive examination of variables took place. This was followed by a bivariate analysis. Each independent variable was individually assessed to discern its relation to
changes in plasma BDNF levels or G-DNA-M percentage (dependent variables). Graphical analysis was incorporated to ascertain the nature of the relationship, be it linear or otherwise.

### Multivariate analysis

To address the proposed changes in plasma BDNF levels and/or G-DNA-M percentage during the final evaluation, we crafted empirical predictive models (refer to Fig. 1). Initially, for numerical variables, we graphically represented them to gauge the data curve's attributes, both for the entire cohort and by gender. This visual exploration revealed that the relationships between the aforementioned predictive variables (changes in final plasma BDNF levels and global DNA methylation percentage) aren't linear (see Fig. 1A). Hence, we employed restricted splines to capture the non-linear interplay between these variables (*Gauthier, Wu & Gooley, 2020*; *Harrell, 2015*). Restricted splines effectively represent a non-linear relationship through a linear one, essentially transforming an independent variable. The independent variable's value range was segmented, with "knots" demarcating one segment's conclusion and the next's commencement. Consequently, splines ensure the resultant curve remains smooth and continuous, minimizing model misrepresentation and elucidating the predictor-outcome relationship (*Gauthier, Wu & Gooley, 2020*).

Figure 1C illustrates the quest for predictive strength ($R^2$), focusing on the change in a variable without a definitive cut-off for a robust model. In the empirical development of a multivariate predictive model, it became evident that variable relationships weren't linear, necessitating the use of restricted splines. Based on one model's Anova, we proposed a subsequent model. This involved conceptually discarding non-pertinent variables and pinpointing those with insignificant linear and non-linear components. We then compared the new model to its predecessor using likelihood ratio tests for two nested models (see Fig. 1B). Relying on the computed adjusted $R^2$ ($R^2aj$), we subjected the empirical model to validation *via* bootstrapping (over 10,000 iterations) as a model validation criterion (*Efron, 1979*; *Harrell, 2015*).

We began with a comprehensive model encompassing 16 predictors for changes in plasma BDNF and G-DNA-M percentage. Using the likelihood ratio test, variables were systematically eliminated until a final model emerged for each metric. This model underwent validation *via* Bootstrap (*Efron, 1979*; *Ledesma, 2008*). We derived the empirical predictive power ($R^2e$), offering a tangible gauge of the model's predictive prowess and future performance. Subsequently, for an average individual, predictions were made based on a specific independent variable, while controlling others. This determined changes in the final *versus* initial evaluations in plasma BDNF or G-DNA-M percentage. Database creation utilized Microsoft Excel 365, while R (*R Core Team, 2023*) facilitated statistical analysis, setting a significance threshold at $\alpha = 0.05$.

Figure 1: Illustrates the multivariate analysis process, the empirical development of a predictive model, and its theoretical foundation. Figure 1A displays the relationship between final physical and psychological reactions and the logarithm of BDNF change. The non-linear nature of these variables prompted the use of restricted splines in our predictive model. These splines effectively transform a non-linear relationship into a linear one. Figure 1B depicts the relationship between final physical and psychological reactions and
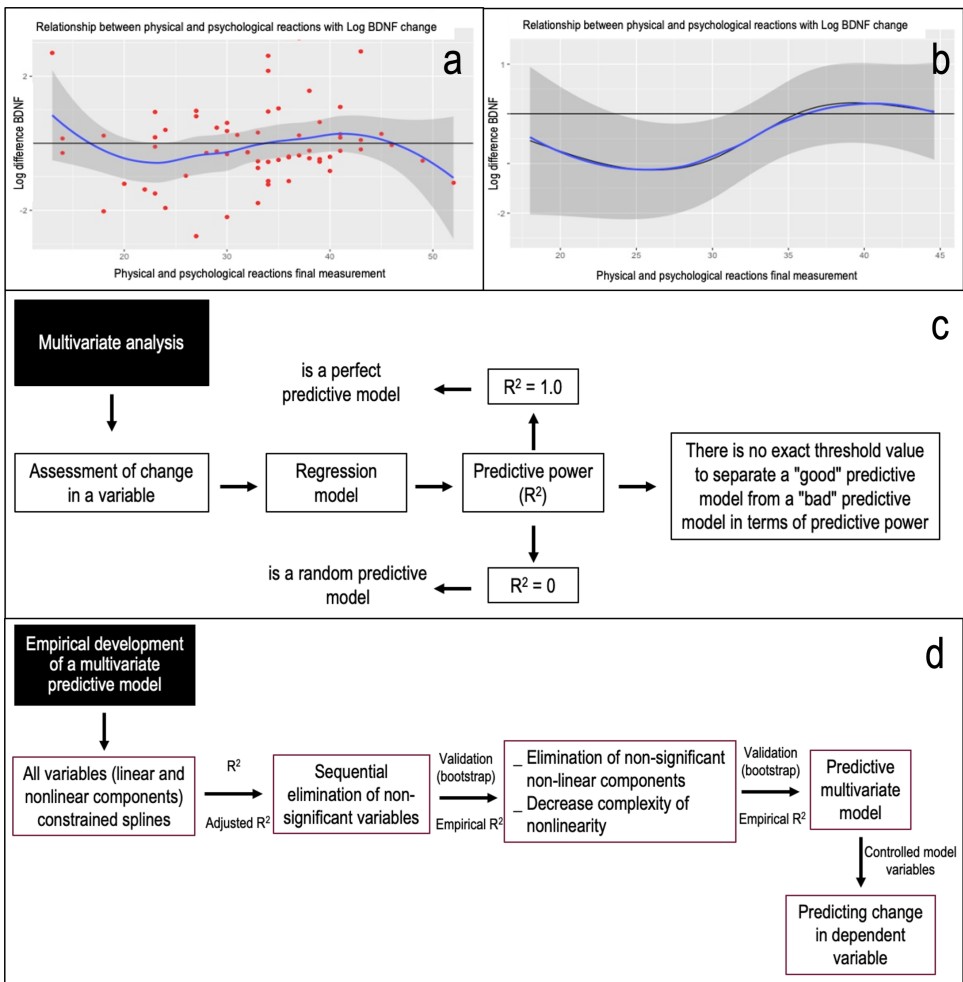

**Figure 1** **The process of multivariate analysis and empirical development of a predictive model and its theoretical reasoning is shown.** In (A) the relationship between physical and psychological reactions in the final assessment with the logarithm of the change in BDNF is shown as an example. It can be seen that there is no linear behavior of the variables, which is why we proceeded to develop a predictive model using restricted splines. These are a ways of summarising a non-linear relationship usefully using a linear relationship, being only a transformation of an independent variable. In (B) the relationship between physical and psychological reactions in the final assessment with the logarithm of the change in BDNF with the variable controlled as a function of the model is shown as an example. When comparing the curve of the variables, with the line of no change, an area not covered by the confidence interval (24 to 31 points) is observed. Thus, in this case "in the population, when there are between 24 and 31 points in physical and psychological reactions in their final assessment, there will be a decrease in the final levels of plasma BDNF with respect to the initial ones." (C) The panel explains the search for predictive power ($R^2$) based on objectifying the change in a variable and the lack of a cut-off value to define a good model. (D) The process of empirical development of a predictive multivariate model. Considering that the relationships between variables do not behave linearly, an analysis for non-linear effects (restricted splines) was considered. Consequently, in iterative form and based on Anova on one model, a next model was proposed by conceptually eliminating non-relevant variables and identifying those variables in which both their linear and non-linear components were not significant and comparing the new model with the previous one, based on likelihood ratio tests for two nested models. Thus, based on calculated predictive power or calculated adjusted $R^2$ ($R^2$aj), the empirical model was subjected to validation based on bootstrapping (over 10,000 iterations) as a criterion for validating the predictive model (*Harrell, 2015*), Obtaining the empirical predictive power ($R^2$e).

the logarithm of BDNF change, with the variable controlled based on the model. Comparing the variable curve to the no-change line reveals a confidence interval gap (24 to 31 points). Figure 1C delves into the quest for predictive power ($R^2$). It emphasizes the challenge of pinpointing a definitive cut-off value for a robust model. Figure 1D outlines the empirical development of a multivariate predictive model. Given the non-linear relationships among variables, the analysis incorporated non-linear effects using restricted splines. In the context of the population, when scores for physical and psychological reactions in the final assessment range between 24 and 31 points, a decline in end-of-study plasma BDNF levels compared to initial levels is anticipated. Using an iterative approach and based on one model Anova, a subsequent model was proposed. This model conceptually removed non-essential variables and pinpointed those with insignificant linear and non-linear components. The new model was then compared to its predecessor using likelihood ratio tests for nested models. With the adjusted $R^2$ ($R^2$aj) as a guide, the empirical model underwent validation through bootstrapping, exceeding 10,000 iterations (*Harrell, 2015*). This process yielded the empirical predictive power ($R^2$e).

# RESULTS

## Sample characteristics

Of the participants, 63.0% (41) were females and 36.92% (24) were males. At baseline, females averaged 21.14 years (range: 19.10–24.53) and males 21.36 years (range: 18.84–28.12). At the final assessment, females averaged 21.95 years (range: 19.89–25.30) and males 22.13 years (range: 19.66–28.85).

## Results for instruments and laboratory evaluations (Table 1)

Significant differences emerged between baseline and final assessments for SISCO-II-AS and Beck Depression Inventory-II. This was evident in stressors score, physical and psychological reactions (Pprx), social behavioural reactions (Sbrx), coping strategies, complete SISCO-II-AS, and BECK-II (Wilcoxon test; $p < 0.001$). Notably, the final Pprx showed significant gender differences (Mann–Whitney–Wilcoxon; $p = 0.020$). The AUDIT test and SRQ inventory, assessed only at the end, revealed no gender differences (Mann–Whitney–Wilcoxon test; $p > 0.05$). Baseline plasma BDNF averaged 1800.43 ($\pm$1957.91) pg/ml, and 1676.94 ($\pm$1805.09) pg/ml at the final assessment. G-DNA-M values were 2.40 ($\pm$1.40) % and 2.57 ($\pm$1.91) %, respectively. Neither variable showed significant changes between the two assessments (Wilcoxon test; $p > 0.05$). All participants were heterozygous (Val/Met genotype), excluding it from the analysis.

## Multivariate analysis
### Predicting BDNF plasma level changes

The comprehensive initial model was significant ($R^2 = 0.187$; $p < 0.001$). Several variables, including baseline stressors, coping, final stressors, final Sbrx, and both baseline and final Beck-II, were removed. This led to a foundational model, which underwent further optimization. Bootstrap methods yielded an empirical $R^2$ ($R^2$e) for the population (*Efron, 1979*). Non-significant non-linear components, such as SRQ20 and baseline G-DNA-M,

**Table 1** **Summary distribution of participants by sex according to SISCO-II inventory of academic stress and its components, Beck-II inventory, plasma BDNF (pg/ml) and percentage of global DNA methylation.** Baseline and final assessment. Non-parametric contrast test.

| Variables | S. | Baseline assessment | | | | Final assessment | | | | p diff. time[**] |
|---|---|---|---|---|---|---|---|---|---|---|
| | | All | Men | Women | p diff sex[*] | All | Men | Women | p diff sex[*] | |
| Stressors | M | 14.43 | 15.83 | 13.61 | | 26.45 | 26.58 | 26.37 | | |
| | SD | 7.77 | 8.31 | 7.42 | 0.246 | 5.58 | 6.10 | 5.33 | 0.653 | <0.001 |
| Physical and psychological reactions | M | 23.91 | 22.12 | 24.95 | | 32.68 | 29.33 | 34.63 | | |
| | SD | 6.63 | 5.33 | 7.13 | 0.151 | 8.42 | 9.75 | 6.94 | 0.020 | <0.001 |
| Social behavioural reactions | M | 11.83 | 12.33 | 11.54 | | 17.09 | 16.42 | 17.49 | | |
| | SD | 3.74 | 3.70 | 3.77 | 0.290 | 4.57 | 4.61 | 4.57 | 0.567 | <0.001 |
| Coping strategies | M | 16.97 | 16.58 | 17.20 | | 19.27 | 18.17 | 19.90 | | |
| | SD | 5.45 | 5.18 | 5.66 | 0.749 | 3.94 | 3.85 | 3.89 | 0.118 | <0.001 |
| SISCO-II-AS | M | 67.14 | 66.88 | 67.29 | | 95.98 | 90.57 | 99.10 | | |
| | SD | 16.09 | 14.86 | 16.95 | 0.962 | 16.66 | 20.00 | 13.72 | 0.139 | <0.001 |
| Beck-II | M | 9.75 | 8.30 | 10.56 | | 17.65 | 14.83 | 19.27 | | |
| | SD | 7.89 | 5.87 | 8.79 | 0.528 | 11.23 | 9.12 | 12.09 | 0.157 | <0.001 |
| Plasma BDNF | M | 1800.43 | 1819.21 | 1789.78 | | 1676.94 | 1377.77 | 1852.06 | | = |
| | SD | 1957.91 | 2393.71 | 1684.78 | 0.471 | 1805.09 | 1646.65 | 1889.22 | 0.221 | 0.258 |
| G-DNA-M | M | 2.40 | 2.24 | 2.50 | 0.369 | 2.57 | 2.68 | 2.50 | 0.086 | = |
| | SD | 1.40 | 1.36 | 1.44 | | 1.91 | 1.23 | 2.22 | | 0.759 |

**Notes.**

S, Statistic; M, arithmetic mean; SD, standard deviation; SISCO-II-AS, SISCO-II Academic Stress Inventory; BDNF, plasma BDNF (pg/ml); G-DNA-M, global DNA methylation percentage.

[*]$p$-value of Mann–Whitney–Wilcoxon test for differences between males and females.

[**]$p$-value of Wilcoxon test for difference between baseline and final measurement.

Maximum possible score Stressors = 40 points; Maximum possible score Physical and psychological reactions = 55 points; Maximum possible score Social behavioural reactions = 30 points; Maximum possible score Total reaction = 85 points; Maximum possible score Coping strategies = 30 points; Maximum possible score SISCO-II Inventory = 155 points; Maximum possible score Beck-II = 63 points.

were then removed. The non-linearity complexity in variables like baseline Pprx, baseline Sbrx, and final G-DNA-M was reduced. The refined predictive model displayed $R^2 = 0.524$; $R^2a = 0.368$; $p < 0.001$, and $R^2e = 0.172$ (Table 2). After finalizing the model, predictions for an average individual's plasma BDNF changes were made. To analyze the effect of a specific variable, the values of other model variables were set to their median. This controlled variable approach nullified the effects of other variables. Table 3 presents the model's baseline with variables at their median. It also displays the average percentage change in plasma BDNF from baseline to the final assessment. The model revealed a 3.56% decrease in females' plasma BDNF at the final assessment compared to baseline. For males, the decrease was 17.14%.
**Table 2 Multivariate model for change in plasma BDNF or in percentage of global DNA methylation at final assessment relative to baseline assessment.** It shows the initial model and the optimised model.

| Variables | Plasma BDNF | | Global DNA methylation | |
|---|---|---|---|---|
| | Initial model | Optimised model | Initial model | Optimised model |
| Str-BA° | −0.048 (0.580) | – | 0.054 (0.458) | – |
| Str-BA'' | 0.138 (0.410) | – | −0.089 (0.509) | – |
| Pprx-BA° | 0.221 (0.296) | 0.220 (0.003) | 0.360 (0.033) | 0.243 (0.002) |
| Pprx-BA'' | −0.191 (0.741) | −0.217 (0.003) | −0.502 (0.234) | −0.184 (0.020) |
| Pprx-BA'''' | −0.125 (0.938) | – | 0.838 (0.456) | – |
| Sbrx-BA° | −0.297 (0.487) | −0.612 (<0.001) | 0.849 (0.059) | 0.497 (0.018) |
| Sbrx-BA'' | −0.305 (0.764) | 0.584 (<0.001) | −2.700 (0.008) | −2.009 (<0.001) |
| Sbrx-BA'''' | 2.642 (0.409) | – | 9.235 (0.003) | 7.207 (<0.001) |
| Coping-BA° | −0.264 (0.205) | – | −0.192 (0.250) | – |
| Coping-BA'' | 0.626 (0.240) | – | 0.368 (0.402) | – |
| Coping-BA'' | −1.457 (0.344) | – | −0.961 (0.448) | – |
| Str-FA° | 0.177 (0.337) | – | −0.091 (0.545) | −0.131 (0.165) |
| Str-FA'' | −0.661 (0.329) | – | 0.937 (0.095) | 1.041 (0.004) |
| Str-FA'''' | 1.784 (0.359) | – | −2.885 (0.078) | −3.241 (0.002) |
| Pprx-FA° | −0.068 (0.616) | −0.140 (0.021) | −0.030 (0.813) | – |
| Pprx-FA'' | 0.382 (0.106) | 0.414 (0.002) | 0.028 (0.905) | – |
| Pprx-FA'''' | −1.984 (0.101) | −2.003 (0.005) | −0.048 (0.968) | – |
| Sbrx-FA° | −0.283 (0.190) | – | 0.206 (0.342) | 0.262 (0.044) |
| Sbrx-FA'' | 0.493 (0.237) | – | −0.638 (0.133) | −0.746 (0.005) |
| Sbrx-FA'''' | −1.994 (0.438) | – | 4.680 (0.073) | 5.030 (0.002) |
| Coping-FA° | 0.329 (0.354) | 0.470 (0.006) | 0.071 (0.780) | −0.043 (0.266 |
| Coping-FA'' | −1.064 (0.295) | −1.420 (0.002) | −0.330 (0.630) | – |
| Coping-FA'''' | 3.646 (0.350) | 4.970 (0.004) | 1.396 (0.606) | – |
| Beck-II-BA° | 0.021 (0.935) | – | −0.387 (0.142) | −0.315 (0.029) |
| Beck-II-BA'' | −0.091 (0.953) | – | 2.051 (0.156) | 1.732 (0.029) |
| Beck-II-BA'''' | 0.117 (0.973) | – | −4.573 (0.146) | −3.951 (0.036) |
| Beck-II-FA° | 0.073 (0.563) | – | 0.050 (0.591) | 0.015 (0.672) |
| Beck-II-FA'' | −0.900 (0.456) | – | −0.576 (0.539) | 0.111 (0.046) |
| Beck-II-FA'''' | 1.790 (0.449) | – | 1.304 (0.476) | – |
| AUDIT° | −0.814 (0.298) | – | 1.129 (0.083) | 0.930 (0.017) |
| AUDIT'' | 8.784 (0.323) | – | −14.42 (0.041) | −13.48 (0.003) |
| AUDIT'' | −14.05 (0.327) | – | 23.30 (0.039) | 22.08 (0.002) |
| SRQ20° | −0.345 (0.294) | −0.116 (0.009) | −0.173 (0.549) | −0.245 (<0.001) |
| SRQ20'' | 0.899 (0.344) | – | 0.261 (0.735) | – |
| SRQ20'''' | −2.166 (0.315) | – | −0.911 (0.603) | – |
| G-DNA-M-BA° | 0.395 (0.691) | 0.441 (<0.001) | – | – |
| G-DNA-M-BA'' | 1.771 (0.722) | – | – | – |
| G-DNA-M-BA'' | −8.693 (0.635) | – | – | – |
| G-DNA-M-FA° | 0.687 (0.597) | 0.563 (0.020) | – | – |
| G-DNA-M-FA'' | −1.083 (0.874) | −0.975 (0.007) | – | – |
| G-DNA-M-FA'''' | 0.774 (0.963) | – | – | – |

**Table 2** (*continued*)

|  | Plasma BDNF | | Global DNA methylation | |
| Variables | Initial model | Optimised model | Initial model | Optimised model |
| --- | --- | --- | --- | --- |
| BDNF-BA° | – | – | −0.002 (0.291) | <0.001 (0.463) |
| BDNF-BA'' | – | – | 0.022 (0.309) | 0.002 (0.103) |
| BDNF-BA'''' | – | – | −0.036 (0.329) | – |
| BDNF-FA° | – | – | −0.002 (0.104) | −0.001 (0.001) |
| BDNF-FA'' | – | – | 0.022 (0.395) | 0.003 (0.003) |
| BDNF-FA'''' | – | – | −0.033 (0.439) | – |
|  |  |  |  |  |
| $R^2$ | 0.776 | 0.524 | 0.787 | 0.686 |
| $R^2$a | 0.187; $p < 0.001$ | 0.368; $p < 0.001$ | 0.226; $p < 0.001$ | 0.431; $p < 0.001$ |
| $R^2$e | – | 0.172 | – | −0.448 |

**Notes.**

Variables: BA, Baseline Assessment; FA, Final Assessment; Str, SISCO-II-AS Stressors; Pprx, SISCO-II-AS Physical and Psychological Reactions; Sbrx, SISCO-II-AS Social Behavioural Reactions; Coping, SISCO-II-AS Coping Strategies; Beck-II, Beck Depression Inventory; SRQ20, Self-Report Symptom Questionnaire; AUDIT, Alcohol Use Disorders Identification Test; BDNF, Plasma BDNF; G-DNA-M, Global DNA methylation percentage.

(°) Linear component of the variable; ('') First node non-linear component of the variable (restricted splines); ('''') Second node non-linear component of the variable (restricted splines). $R^2$: Coefficient of determination. $R^2$a: Adjusted coefficient of determination. $R^2$e: Empirical coefficient of determination, obtained by bootstrapping over 10,000 iterations. Significance level of $\alpha = 0.05$.

**Table 3** The mean percentage change at final assessment from baseline assessment, as a function of a given variable and with all other variables controlled. Predicted percentage change in plasma BDNF levels and percentage of global DNA methylation.

| Considered variable | Controlled variables (score)[*] | Average percentage change of final evaluation compared to initial evaluation | |
| --- | --- | --- | --- |
|  |  | Women | Men |
| **Plasma BDNF** | Pprx-BA (24); Sbrx-BA (11); Pprx-FA (34); Coping-FA (20); SRQ20 (10); G-DNA-M-BA (2.22+); G-DNA-M-FA (2.06+). | −3.56 | −17.14 |
| **Global DNA methylation** | Beck-II-BA (8); Beck-II-FA (16); Pprx-BA (24); Sbrx-BA (11); Str-FA (27); Sbrx-FA (17); Coping-FA (20); AUDIT (2); SRQ20 (10); BDNF-BA (951.5+); BDNF-FA (960.30+). | 15.06 | 18.96 |

**Notes.**

[*]Reference point of the model with variables fixed at their median.

Variables: BA, Baseline Assessment; FA, Final Assessment; Str, Stressors SISCO-II-AS; Pprx, Physical and Psychological Reactions SISCO-II-AS; Sbrx, Social Behavioural Reactions SISCO-II-AS; Coping, Coping Strategies SISCO-II-AS; Beck-II, Beck Depression Inventory; SRQ20, Self-Report Symptom Questionnaire; AUDIT, Alcohol Use Disorders Identification Test; BDNF, Plasma BDNF; G-DNA-M, Global DNA methylation percentage.

***For predicting changes in global DNA methylation percentage***

The initial model, encompassing all variables, proved significant ($R^2 = 0.226$; $p < 0.001$). After removing baseline stressors, baseline coping, and final Pprx, a preliminary model emerged. Further optimization removed non-significant non-linear components: final coping and SRQ20. Non-linearity complexity was then reduced to three points for variables Pprx-EB, final Beck-II, baseline BDNF, and final BDNF. This yielded the simplest predictive model ($R^2 = 0.686$; $R^2$a $= 0.431$; $p < 0.001$, $R^2$e $= -0.448$) (Table 2). To forecast G-DNA-M percentage changes at the final assessment compared to the baseline, we set values for the other model variables. This was done when examining the impact of a specific variable.

Table 3 presents the baseline of the model from which predictions were made. In females, the G-DNA-M percentage at the final assessment increased, on average, by 15.06% from the baseline. In males, the increase was 18.96%.

## DISCUSSION

In our quest to understand the relationship between AS and its potential impact on plasma BDNF and G-DNA-M percentage, we embarked on the creation of empirical models. Although there are overarching guidelines for such modelling, the specificity of our predictive objectives called for a distinct data set. The construction and interpretation of these models were driven by a blend of statistical rigour and theoretical understanding.

To investigate the link between AS and plasma BDNF shifts, we employed a 16-variable model. This model showed clear significance ($R^2a = 0.187$; $p < 0.001$). Later, seven non-impactful variables were discarded. One anticipated exclusion was the variable representing excessive alcohol consumption (AUDIT) from the final evaluation. The average participant score ($2.88 \pm 2.86$ points) was beneath the 6–8 point risk consumption threshold (*Alvarado et al., 2009*; *Donoso, 2015*). Previous studies have underscored alcohol's influence on peripheral BDNF levels (*Umene-Nakano et al., 2009*). It's noteworthy that while serum BDNF levels experience an uptick 14 days after halting alcohol consumption, they don't parallel the levels seen in non-drinking counterparts even after half a year (*Girard et al., 2020*; *Heberlein et al., 2016*). Notably, BDNF levels in adolescents demonstrated a decline among those who consumed alcohol (*Miguez et al., 2020*).

The variables from the Beck-II depression inventory, both final and baseline, were excluded. Despite a significant rise in the Beck-II score at the final assessment ($p < 0.001$), the average was 17.65 (11.23) points, under the 18-point threshold. This evaluation occurred amidst academic stress, potentially highlighting undetected depressive symptoms from the baseline.

Peripheral BDNF decrease is noted in major depressive disorder patients. Various therapies, including pharmacological and psychotherapeutic, only achieve partial BDNF increase (*Aldoghachi et al., 2019*; *Bocchio-Chiavetto et al., 2010*). They don't match levels in healthy individuals (*Brunoni, Lopes & Fregni, 2008*; *Kishi et al., 2018*). Studies using Beck-II find a BDNF-depressive symptomatology link (*Cheon et al., 2018*; *Just, Fraszczak & Araszkiewicz, 2016*). Male university students with the disorder showed reduced plasma BDNF levels. Their levels were measured using Beck-II, without linking to academic overload (*Tavakoli et al., 2017*).

In the SISCO-II-AS framework, stressors were excluded from both initial and concluding assessments. Stressors encapsulate scenarios students perceive as academic pressures (*Castillo-Navarrete et al., 2020*). A comparison between the assessments revealed discernible differences (Table 1). This, combined with other SISCO-II-AS elements, indicates the manifestation of AS in the study's latter phase.

In the SISCO-II-AS model, Pprx's importance is clear. In the final assessment, it rose significantly ($p < 0.001$). Items in it suggest self-regulation loss. Stress affects neurotrophin expression, especially BDNF (*Bath, Schilit & Lee, 2013*). Stress outcomes are shaped by

form, duration, timing, and the individual. The HPA axis is activated by psychological stress, amplifying cortisol levels while diminishing BDNF (*Brunoni, Lopes & Fregni, 2008*). Sexual dimorphism in stress responses, notably in BDNF levels, showed pronounced variations: men at 17.14% and women at 3.56%. BDNF's interplay with cortisol during acute stress suggests assessing peripheral BDNF can shed light on HPA axis operations (*Linz et al., 2019*). Persistent stress impacts the HPA axis, lowering BDNF (*Miller, Chen & Zhou, 2007*; *Schmitt, Holsboer-Trachsler & Eckert, 2016*; *Schulte-Herbrüggen et al., 2006*). This axis plays a role in initiating and concluding sleep (*Han, Kim & Shim, 2012*). The mention of sleep disorders in Pprx emphasizes sleep's importance in cognitive processes. Predictably, individuals with insomnia have reduced BDNF levels, with a direct correlation to the severity of their insomnia (*Elliott et al., 2014*; *Grønli, Soulé & Bramham, 2014*; *Schmitt, Holsboer-Trachsler & Eckert, 2016*).

Based on the SISCO-EA's AS construct, responses emphasize stressors' effects, not causes. Responses, whether conscious or unconscious, are physiological and psychological. This amplifies the total reaction (Pprx and Sbrx), indicating regulatory capacity loss and disrupted homeostasis (*Castillo-Navarrete et al., 2020*). Furthermore, coping strategies rise in reaction to stressor consequences. This is evident as Pprx and coping variables increased significantly in the final assessment.

The constant intellectual challenge posed by university studies provides ongoing cognitive stimulation. Such cognitive training has the potential to counteract the BDNF-reducing effects of academic stress, thereby potentially maintaining or even increasing peripheral BDNF levels (*Al-Thaqib et al., 2018*; *Angelucci et al., 2016*; *Damirchi, Hosseini & Babaei, 2018*; *Jahangiri, Gholamnezhad & Hosseini, 2019*; *Kallies et al., 2019*; *Ledreux et al., 2019*; *Miyamoto et al., 2018*). Alongside this, the practice of hobbies, particularly those that involve physical activity, can enhance BDNF levels, offering another potential buffer against academic stress (*Al-Thaqib et al., 2018*; *Damirchi, Hosseini & Babaei, 2018*; *Jahangiri, Gholamnezhad & Hosseini, 2019*; *Kallies et al., 2019*; *Ledreux et al., 2019*; *Miyamoto et al., 2018*).

The percentage of G-DNA-M showed no significant differences between baseline and final assessments. Thus, linking these variables to plasma BDNF change remains uncertain. In African-Americans, increased G-DNA-M was observed in post-traumatic stress individuals (*Smith et al., 2011*). This suggests psychosocial stress may alter G-DNA-M patterns, affecting perception and physiological responses to stressors. Since G-DNA-M state reduction inhibits transcription, a potential BDNF level decline could result from heightened DNA methylation. To clarify this, specific BDNF gene methylation and related genes should be examined.

By removing non-linear components and simplifying non-linearity, we derived the most straightforward predictive model. This validated model ($R^2$aj $=0.368$; $R^2$e $=0.172$) applies to populations under evaluated conditions and procedures. It predicts future BDNF plasma levels under these conditions. This model aids in predicting plasma BDNF changes at the final assessment, partially addressing the study's aim. Predictions indicated a pronounced sex difference in plasma BDNF changes: women's plasma BDNF decreased by 3.56% from baseline at the final assessment, while men's showed a more considerable decrease of

17.14% (Table 3). This disparity underscores the need to further explore the role of sex in modulating BDNF responses to academic stress.

We empirically crafted a multivariate model predicting G-DNA-M level changes at final assessments. The initial model, comprising 16 variables, showed notable predictive capacity ($R^2a = 0.226$; $p < 0.001$). Variables like baseline stressors, baseline coping, and final Pprx were then discarded. Intriguingly, the final Pprx, a primary AS indicator and a strong self-regulation loss signal, wasn't significant. This might suggest an inadequate sample size or that chronic stress doesn't impact G-DNA-M levels. However, these propositions require validation through dedicated studies.

Sexual dimorphism in stress response has been well-documented, and our findings support these established notions. Besides the observed differences in G-DNA-M between genders, it is essential to highlight the disparity in BDNF changes between men and women. The pronounced decline in men's BDNF levels compared to women suggests an inherent difference in how chronic academic stress impacts the neurotrophic support system between genders (*Castillo-Navarrete et al., 2023b*). This divergence, especially within the context of the HPA axis activity, provides a nuanced layer to the understanding of the relationship between stress, G-DNA-M, BDNF, and gender. Such findings mandate a deeper exploration of sex-specific mechanisms and interventions in managing academic stress.

Significant variables in this intricate model are final Beck-II and baseline Beck-II. Global DNA hypomethylation is documented in major depressive disorder (*Bakusic et al., 2017*; *Nantharat et al., 2015*). Additionally, some studies link major depressive disorder to stressful events (*Bakusic et al., 2017*; *Duman & Canli, 2015*). Early life stresses and psychosocial factors are central to these analyses (*Bakusic et al., 2017*; *Duman & Canli, 2015*; *Nantharat et al., 2015*; *Tseng et al., 2014*; *Unternaehrer & Meinlschmidt, 2016*).

Psychosocial stress might impact specific genes' G-DNA-M patterns. These genes concern individual physiological responses and conditioning factors (*Lisoway et al., 2019*; *Smith et al., 2011*; *Wolf et al., 2018*). A stressor's presence activates the HPA axis. This activation encompasses both the stimulatory pathways and the feedback loops which are involved in regulating the production of stress hormones, including cortisol (*Miller, Chen & Zhou, 2007*; *Schmitt, Holsboer-Trachsler & Eckert, 2016*; *Schulte-Herbrüggen et al., 2006*). About post-traumatic stress, an increase in G-DNA-M has been documented (*Archer et al., 2019*; *Mehta et al., 2020*; *Notaras & Van den Buuse, 2020*; *Roth et al., 2011*; *Smith et al., 2011*). For a more precise understanding, studies focusing on the specific methylation of particular genes associated with this system would be beneficial.

While our study focused on G-DNA-M and BDNF as indicators of stress response, we acknowledge the absence of direct cortisol measurements. Cortisol, a primary stress hormone, would offer immediate insights into HPA axis activity during stress events. Future studies should incorporate cortisol levels to provide a comprehensive assessment of the stress response.

The model, after validation and refinement, showed a negative $R^2e$ ($-0.448$), raising questions about its prediction accuracy. In our sample, women's G-DNA-M percentage rose by 15.06% in the final check. For men, it was an 18.96% rise (Table 3). The SISCO-II-AS
model predicts a G-DNA-M percentage increase in the final assessment. While these variables suggest the AS phenomenon, the influence of unidentified variables remains uncertain. They might activate the HPA axis and other stress-regulating neuroendocrine systems. These could also cause specific and global methylation changes (*Archer et al., 2019*; *Bakusic et al., 2017*; *Duman & Canli, 2015*; *Lisoway et al., 2019*).

In this study, a rigorous approach was used. We employed psychometric variables with a validated history and an explicit application method, which is a strength of this study. An adjusted $R^2$ was obtained, suggesting the possibility of extrapolating the results to a broader population. However, the robustness of the study is potentially limited by the small sample size. Furthermore, it should be noted that our sample consisted only of students with strong biological and scientific backgrounds. This could significantly influence how the phenomenon of AS is manifested and managed, possibly biasing the overall applicability of the results. This specificity in the sample represents both a strength, in terms of its well-defined scope, and a limitation, given the potential challenges in broader applicability.

Regarding plasma BDNF levels, the study recognizes certain limitations. Several pre-analytical variables were not accounted for. These exclusions extend not only to the individual participants' characteristics but also to the technical aspects like the plasma separation process after obtaining the peripheral blood sample.

The decision to use the percentage determination of G-DNA-M has its limitations, underscoring the study's cautious approach. Drawing expansive conclusions from this measure is restrained. A more comprehensive understanding would necessitate specific methylation studies targeting the promoter level of different BDNF gene exons and genes associated with the HPA axis function. This represents an area for potential improvement and further inquiry.

While we've examined global DNA methylation, we acknowledge the absence of specific BDNF promoter methylation analyses. Indeed, examining DNA methylation at BDNF gene promoters would provide more nuanced insights into BDNF regulation (reference to BDNF promoter studies). We concede that correlating global methylation with BDNF levels is a preliminary approach. It is crucial to remember that BDNF regulation is multifaceted, encompassing DNA methylation, histone modification, and non-coding RNA interactions. Our current findings only touch on a portion of these intricate regulatory mechanisms. In future projects, we aim to delve deeper into BDNF's epigenetic regulation. Understanding how academic stress impacts BDNF through epigenetics, especially DNA methylation, is paramount.

From the developed predictive models, a proposal emerges. AS may relate to plasma BDNF level changes in the wider population. Also, it could correlate with G-DNA-M percentage shifts in our analyzed sample, considering the noted limitations. Given these findings, future endeavours should validate the model's efficacy for predicting final BDNF levels in an expanded cohort of university students, specifically its link with AS.

There is a pressing and undeniable need to delve deeper into the intricacies of AS and its potential impacts on genetic architecture. Specifically, meticulous studies focusing on the methylation patterns at the exon level of the BDNF gene are paramount. Recognizing the critical importance of direct measurements of cortisol, we have initiated a new project

assessing peripheral blood stress conditions and salivary cortisol levels in university students with varying degrees of academic stress. Preliminary protocols and methodologies for this study have been outlined (*Castillo-Navarrete et al., 2023a*). Equally vital is the investigation of genes intricately linked with the hypothalamic-pituitary-adrenal (HPA) axis. Such granularity in analysis can uncover nuanced mechanisms of stress response and regulation. Comprehensive examinations encompassing both genetic markers and cortisol determinations will be crucial in furthering our understanding of the AS phenomenon.

Study Limitations: In this investigation, certain limitations must be acknowledged. Primarily, our study utilized a convenience sample, which might not be fully representative of all university students. This sampling method could introduce biases that might influence our findings' generalizability. Additionally, our reliance on self-report scales can present challenges. While these scales have been validated and widely used, there is always a possibility of participant bias, recall errors, or misinterpretation of questions, potentially affecting the accuracy of the data.

Our findings provide preliminary insights into the neurobiological markers associated with academic stress among university students. Recognizing the potential relationship between academic stress, plasma BDNF levels, and G-DNA-M percentages could offer new avenues for early intervention and support. Educational institutions, in collaboration with healthcare providers, might consider incorporating strategies such as regular mental health screenings, stress management workshops, and promoting activities known to elevate BDNF levels, like exercise and cognitive training. Such measures could proactively address academic stress, fostering a holistic well-being environment for students.

While our focus has been on academic stress and its influence on BDNF levels, we acknowledge the multifaceted nature of BDNF regulation. Other protective factors such as diet, social interactions, and mental well-being practices undertaken by some students might also play a role in BDNF dynamics. Genetic predispositions, where certain individuals might possess inherent resilience to stress-induced BDNF changes, should also be considered. Further studies investigating these additional factors in conjunction with academic stress would provide a more comprehensive understanding of BDNF regulation in this context.

## CONCLUSIONS

Our study revealed notable differences between the initial and final assessments concerning SISCO-II-AS and its components. This underscores the presence of academic stress in the concluding evaluation. Yet, there were no discernible differences in terms of plasma BDNF and G-DNA-M percentages. We constructed and validated multivariate models, considering the SISCO-II-AS-derived variables, and acknowledging the inherent limitations of our approach. The predictive model focused on final plasma BDNF levels. It indicated a decline in the average final plasma BDNF percentage for both genders. This observation spanned the broader population. Conversely, another model projected G-DNA-M percentage fluctuations by the study's end. It foresaw an increase in average G-DNA-M percentage change for our sample's men and women. Taking study constraints into account, our data offers initial insights. There is a possible connection between

academic stress and plasma BDNF level changes. Similarly, There is a link with G-DNA-M percentage changes among university students.

## ACKNOWLEDGEMENTS

(i) We would like to thank our study participants for their involvement. (ii) Some sections of this article were written with the help of the GPT-4 AI model. A document detailing the Procedure used for the improvement of writing in English is available as supplementary material. However, the results of this study are presented clearly, honestly, and without fabrication, falsification, or inappropriate data manipulation.

### Funding

This work was supported by Grants: Juan-Luis Castillo-Navarrete was supported by OT:2016-6 Facultad de Medicina Universidad de Concepción and by Doctoral Scholarship Conicyt No 21160620. Alejandra Guzman-Castillo was supported by Regular DIN DIREG01/2021, Universidad Católica de la Santísima Concepción. The funders had no role in study design, data collection and analysis, decision to publish, or preparation of the manuscript.

### Grant Disclosures

The following grant information was disclosed by the authors:
OT:2016-6 Facultad de Medicina Universidad de Concepción and by Doctoral Scholarship Conicyt: 21160620.
Regular DIN DIREG01/2021, Universidad Católica de la Santísima Concepción.

### Competing Interests

The authors declare there are no competing interests.

### Author Contributions

- Juan-Luis Castillo-Navarrete conceived and designed the experiments, performed the experiments, analyzed the data, prepared figures and/or tables, authored or reviewed drafts of the article, and approved the final draft.
- Claudio Bustos conceived and designed the experiments, analyzed the data, prepared figures and/or tables, authored or reviewed drafts of the article, and approved the final draft.
- Alejandra Guzman-Castillo performed the experiments, prepared figures and/or tables, authored or reviewed drafts of the article, and approved the final draft.
- Benjamin Vicente conceived and designed the experiments, authored or reviewed drafts of the article, and approved the final draft.

### Human Ethics

The following information was supplied relating to ethical approvals (i.e., approving body and any reference numbers):

All procedures involving human subjects/patients were approved by Scientific Ethical Committee of the Faculty of Medicine of the Universidad de Concepción (NᵒCEC 65/2018)

## Data Availability

The data and codebook are available at the Repositorio de Datos - UdeC: Castillo-Navarrete, Juan-Luis; Bustos, Claudio; Guzman-Castillo, Alejandra; Vicente, Benjamín, 2023, "Increased academic stress is associated with decreased plasma BDNF in Chilean college students", Available at https://doi.org/10.48665/udec/ORVXAA, Repositorio de Datos - UdeC, V1.

## Supplemental Information

Supplemental information for this article can be found online at http://dx.doi.org/10.7717/peerj.16357#supplemental-information.

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
