# Peer review of "Increased academic stress is associated with decreased plasma BDNF in Chilean college students"

_PeerJ, doi:10.7717/peerj.16357_

## Round 0.1 · original submission · Major Revisions

Thank you for submitting your manuscript to the PeerJ. We very much appreciate the time and effort that has gone into the preparation of this article. However, we regret to say that in its current form it is not considered suitable for publication in the PeerJ. The reasons for this decision can be found in the enclosed reports. In general you will see that the reviewers felt positively about the subject matter, however one of them found significant omissions detailed below. If however you feel that you could address all of the points raised then we would be willing to consider a revised manuscript. If you choose this route you should provide a full and detailed rebuttal to all points raised.

Reviewer 1 ·

Basic reporting

1. The authors demonstrate the difference between chronic stress and acute stress, physical stress and psychological stress, and sex differences. But in the discussion part it only contains the sex differences of G-DNA-M, not includes the BDNF (after all, it seems that the sex difference of BDNF predicted by the model is quite obvious: men 3.56% vs women 17.14%). To elaborate more about the sex difference of stress influence on G-DNA-M, BDNF and HPA axis can strengthen the hypothesis of their relationship.

2. In Table 2 (including the legend), the BDNF was mistakenly written as “BNF”.

Experimental design

The authors imply that AS may affect the function of HPA axis, and also mention that chronic stress can increase the expression of cortisol, so why didn’t show the measurement of cortisol level in blood sample before and after the stress. Considering that changes of cortisol level can directly reflect the influence of stress on HPA axis system, I think this result should be provided and discussed.

Validity of the findings

The authors respectively analyze the changes of peripheral blood plasma BDNF and global DNA methylation levels for those college students who suffered AS and imply that the decreased BDNF may be induced by the DNA methylation. In my opinion, the current data is too weak to support this correlation. As it states in the introduction, BDNF can be regulated by not only DNA methylation but the histone modification and non-coding RNA etc. As they mention in the discussion part, to measure the DNA methylation level of promoter region of BDNF is necessary for this study (at least some exons of BDNF transcripts). Without this data, the importance of this study would be greatly reduced. Besides this, I think it should introduce more about how stress affects the BDNF through epigenetics especially the DNA methylation, not simply states the influence of stress on them separately (stress on BDNF or stress on DNA methylation).

Additional comments

In this manuscript the authors assess the influence of academic stress (AS) on college students by inventory and measure the changes of peripheral plasma BDNF level, which has been demonstrated to reflect the brain BDNF level and stress-related mental disorders. In addition, the authors also try to explore the possible mechanisms of BDNF alterations by measuring the percentage of global DNA methylation (G-DNA-M) in peripheral blood. Although there is no significant difference for these two indices between the initial and final assessments, they utilize the multivariate model to further predict the possible changes of peripheral blood plasma BDNF and global DNA methylation after academic stress. Then the authors consider the possible solution for the stress (intellectual challenge of university studies). Overall, the manuscript is well written and mostly quite clear. However, I still have some concerns.

·

Basic reporting

In the present study, the Authors aimed to determine whether academic stress might produce changes in plasma BDNF levels and in the percentage of global DNA methylation. In Authors’ opinion, that I share, this would provide relevant information for the understanding of the phenomenon of academic stress, from a comprehensive view of the network of regulatory mechanisms.
Overall, I found this study very interesting, original, well-conducted and scientifically sound: it adds something new to existing literature on stress and neurotrophins.
As well, I have some minor suggestions aimed at improving the high quality of the paper, and these are described below:
1) In the introduction, a brief note on the fact that BDNF might be influences by several types of stress and can be altered in several psychiatric disorders, but it can be improved by therapeutic treatments, should be added with appropriate references (see doi: 10.1093/ijnp/pyw003 and Martinotti et al. J Biol Regul Homeost Agents. 2012;26(3):347-56).
2) How the participants were recruited? Beside, I believe that isn’t clear whether they were consecutive or randomly selected? And how many subjects were screened, but refused to participate?
3) Was also the presence of a mild intellectual disability evaluated, how, and used as an exclusion criterion? More in detail, what were the exclusion criteria? As well was any recompense provided for the participation in the present study?
4) The Authors should add a separate paragraph on study limitations in the discussion, including the use of a convenience sample and the use of self report scales.
5) Translating into everyday “real world” practice, what possible clinical shreds of evidence might arise from the present study on a clinical point of view and what the Authors do recommend to improve practice and ameliorate academic stress?

Experimental design

Please, see above.

Validity of the findings

Please, see above.

Additional comments

Please, see above.

Reviewer 3 ·

Basic reporting

The study are basically descriptive not mechanical analysis. Overall, the style and language need to be improved in readability and accuracy. For example:
1. The reference style is not consistent in the sections of Main text and References;
2. The references in line 105-106, 113-114 were inserted repeatedly.
3. Two “Participants” in line 150.
4. “….been fully anonymize…” in line 165.
5. In line 197, which does “01 tube” mean?
6. In line 220-221, “…(curve slope) x (DNA concentration in the well)] x 100…” should be “…(curve slope) ×(DNA concentration in the well)] ×100…”
7. In line 369 “…found the between baseline and final assessments…”?
8. In table 2, what does “plasma BNF” mean? Is this "plasma BDNF"?

Experimental design

No comment

Validity of the findings

1. The author reported that changes in the plasma BDNF and global DNA methylation levels between baseline and final assessments in college students in response to academic stress. Whether the DNA methylation level was changed in the promoter of BDNF genes?

2. A significant difference was reported between men and women in the changes of BDNF level in response to academic stress. Please provide the potential explanation.

---

## Round 0.2 · accepted · Accept

Many thanks for addressing all the issues.

Reviewer 1 ·

Basic reporting

no comment

Experimental design

no comment

Validity of the findings

no comment